# Hybrid Magnetic–Inductive Angular Sensor with 360° Range and Stray-Field Immunity

**DOI:** 10.3390/s22062153

**Published:** 2022-03-10

**Authors:** Bruno Brajon, Lorenzo Lugani, Gael Close

**Affiliations:** Melexis Technologies SA, Chemin de Buchaux 38, CH-2022 Bevaix, Switzerland; bur@melexis.com (B.B.); llg@melexis.com (L.L.)

**Keywords:** angle sensor, magnetic, inductive, Hall effect

## Abstract

Magnetic and inductive sensors are the dominant technologies in angular position sensing for automotive applications. This paper introduces a new angular sensor: a hybrid concept combining the magnetic Hall and inductive principles. A magnetic Hall transducer provides an accurate angle from 0° to 180°, whereas an inductive transducer provides a coarse angle from 0° to 360°. For this novel concept, a hybrid target with a magnetic and inductive signature is also needed. Using the two principles at the same time enables superior performances, in terms of range, compactness and robustness, that are not possible when used separately. We realized and characterized a prototype. The prototype achieves a 360° range, has a high accuracy and is robust against mechanical misalignments, stray fields and stray metals. The measurement results demonstrate that the two sensing principles are completely independent, thereby opening the doors for hybrid optimum magnetic–inductive designs beyond the usual trade-offs (range vs. resolution, size vs. robustness to misalignment).

## 1. Introduction

Inductive [1] and magnetic sensors [2] are the favored choices in the automotive world for applications requiring position information. Examples of such applications include accelerator and brake pedals, throttle valves, ride height sensors, turbo actuators, coolant valves, steering angle sensors, parking pawl sensors and motor position sensors, to name a few. What makes them the solution of choice for these applications is their ability to cope with the harsh automotive environments. Dust, oil and metal particle residues, strong humidity excursions, aging and an operational temperature range that can span from −40 °C up to 160 °C are among the principal challenges that these sensors, unlike their capacitive and optical counterparts, can cope with very effectively.

An important sub-family of magnetic sensors are the magneto-resistive (xMR) sensors. The most popular type of xMR sensors in automotive environments is the anisotropic magneto-resistive (AMR) sensor [2]. AMR sensors provide high sensitivity and low hysteresis [3]. However, they are corrupted by magnetic stray fields in a non-linear fashion, greatly complicating the rejection of such disturbances [4].

In recent times, due to the increasing electrification of vehicles, stray-field immunity (SFI) has become a key requirement for automotive sensors [5]. A typical source of stray fields in hybrid or electric cars is the high-current wires, which can produce magnetic fields in the order of several millitesla [6]. A popular solution to stray fields for any sensor that suffers from these disturbances is to magnetically shield the system. This option is bulky and shifts the problem to the sensor integrators [7]. Hall-based magnetic sensors are also traditionally very sensitive to stray magnetic fields. Given their linearity, the stray fields can be readily rejected with a differential scheme [8], thereby meeting the SFI requirement specified in *ISO 11452-8* [9]. This scheme is also referred to as “gradiometric”, as it senses the magnetic field gradient in the plane of the sensor. Such a gradient is generated by a four-pole magnet. This scheme achieves a high accuracy and robustness against mechanical misalignments. The downside is that the range is limited to 180° due to the symmetry of the magnet. However, some applications, such as coolant valves, require sensing over the entire 360° range [10].

Unlike magnetic sensors, inductive sensors operate with AC fields in the MHz range. They are then naturally immune to low-frequency stray fields. Moreover, they are capable of having a very high accuracy [11]. They are bulkier than their magnetic counterparts due to the size of the coil system, which is typically several tens of millimeters large. Given the above limitations, we propose in this paper a hybrid Hall-inductive sensor.

This paper is organized as follows. First, the complementary characteristics of magnetic and inductive sensors are reviewed. The new hybrid concept, which builds on the strengths of each technology, is then described, along with the fabrication of a prototype. Next, the measurement setup and test results are presented, covering all key performance parameters. These results demonstrate that the hybrid sensor successfully extends the range of the magnetic sensor up to a 360° range, maintaining its high accuracy, its robustness against mechanical misalignments and its stray-field immunity without any cross-talk with the inductive sensor. The miniaturization potential is then discussed. The paper concludes with a comparison against other sensors in the state of the art and a discussion of the design perspectives.

## 2. Magnetic Sensor

Figure 1 illustrates the working principle of the magnetic sensor used in the prototype of the hybrid sensor. A four-pole magnet is used to generate a field gradient in the XY plane of the sensor. The field is sensed at eight locations by eight horizontal Hall elements. Compared to vertical Hall elements, these horizontal Hall elements can only detect the out-of-plane magnetic field ($B_{z}$), but they provide a higher sensitivity and lower offset [12]. An integrated magneto-concentrator (IMC), consisting of a high permeability ferromagnetic layer, is used to concentrate and bend the in-plane field components into out-of-plane components ($B_{z}$), which are transduced by the Hall elements. The concentration of the field lines provides a gain factor that can be higher than five in some designs [12]. The Hall elements are combined in two groups of four: the even and the odd group. This results in two quadrature signals related to the field gradients $\delta B_{x}/\delta x-\delta B_{y}/\delta y$ and $\delta B_{x}/\delta y+\delta B_{y}/\delta x$. This scheme is then immune to uniform magnetic stray fields, thereby lowering the angle error in the presence of stray fields. Due to the symmetry of the magnet, the range of the sensor is only 180°. In other words, we have traded the range for improved accuracy.

At first look, it might be tempting to increase the periodicity of the magnetic sensor with a higher-order magnet (with six poles, eight poles, etc.). However, a strong trade-off appears. As the number of poles of the magnet increases, the magnetic field decays faster with the airgap. An increasingly large dead zone, free of the magnetic field, appears under the magnet center. The magnetic signal strength then rapidly deteriorates. To maintain a suitable signal-to-noise ratio under the magnet center, we cannot go beyond a four-pole magnet. The usage of higher-order magnets (up to a 32-pole pair) is possible, but requires a totally different concept [13]: the sensor needs to be placed deliberately off-axis under the poles to recover the signal strength. This off-axis multi-pole arrangement requires more costly magnets and a much tighter mechanical assembly tolerance (in the order of tens of micrometers). These new constraints are not justified, apart from niche applications where angle accuracy is critical and assembly tolerances are tightly controlled (e.g., industrial robots with long rigid arms).

## 3. Inductive Sensor

Figure 2 illustrates the inductive sensing principle. The sensor generates an AC magnetic field via a transmitter coil (Tx). The generated field induces eddy currents into the metallic target. For a 360° range inductive sensor, the metallic target is shaped like a half-moon. The eddy currents inside the metal target locally decrease the magnetic field in accordance with Faraday’s law of induction. The resulting field pattern depends on the target angle. The fields are typically measured by three receiving (Rx) coils. The resulting three-phase voltages induced in the Rx coils encode the angular position of the target.

The strength of the signals detected in the receiving coils plays a major role in the accuracy of the sensor. This is why, in order to obtain a higher signal from the coils, their dimensions need to be in the order of tens of millimeters. This prevents the integration of such a system in small packages. Instead, the coils are implemented on the printed circuit board.

The accuracy of the system is dictated largely by the geometrical properties of the sensor. It is therefore very robust against temperature drifts. Stray metals, on the other hand, are interferers. The eddy currents generated in them influence the fields in the receiving coils. Consequently, the presence of stray metals introduces an angle error.

In general, 360°-range inductive sensors are much more sensitive to mechanical misalignment than higher periodicity inductive sensors. Figure 3 illustrates this effect. It depicts a configuration where the target is misaligned slightly: its center of rotation is shifted to the right. As the target rotates, the eddy current centroid is shifted from its ideal direction, introducing an angular error. The angle error exhibits one period per turn. Consider now a 120°-range inductive sensor for which the metallic target is shaped like a three-lobe clover. The eddy currents in each lobe are shifted like before, but there is some compensation between the lobes. The resulting angle error is significantly reduced. A similar reasoning applies in the case of tilt.

## 4. Hybrid Sensor

Based on the previous discussion, both magnetic and inductive sensors suffer from limitations. They cannot satisfy the ideal set of requirements summarized in Table 1.

For both technologies, the key trade-off is between the accuracy and the range. The accuracy can be readily improved by using high-order symmetrical targets, such as a three-lobe metallic target in inductive sensors (or even a higher number of lobes), or a four-pole magnet in magnetic sensors. The high-order periodicity enables mechanical error compensation by different parts of the sensor (this is true for off-axis and tilt). The periodicity also compresses the electrical errors into a smaller mechanical range: the accuracy, expressed in mechanical degrees, improves, but at the direct expense of the range (the relative accuracy is fixed).

To resolve this trade-off, we propose a hybrid approach, which is depicted in Figure 4. The magnetic sensor is used together with a four-pole magnet. The higher periodicity of the magnet is readily distinguished from the magnetic stray field, and electrical errors are compressed into the 180° range, offering optimum accuracy. To recover the full 360° range, a complementary 360° inductive sensor is used to provide a coarse angle measurement. This coarse measurement is free of the above trade-off, as the accuracy can be greatly relaxed, as this measurement is only used to discriminate between the two 180° sectors. The following equation captures the angle calculation performed by the hybrid sensor: (1)$$\alpha_{hyb}=\left\{\begin{array}{ccc} \alpha_{mag} & \mathrm{when} & \left|\mathrm{angdiff}(\alpha_{mag}-\alpha_{ind})\right|<90{}^{\circ}\\ \alpha_{mag}+180{}^{\circ} & \mathrm{elsewhere} \end{array}\right.$$
where the function angdiff denotes the angular difference wrapped to the interval [−180°, 180°]. The equation makes it clear that the inductive angle $\alpha_{ind}$ can deviate largely without any impact. The deviation, as opposed to the plain angle, appears in a coarse comparison condition with respect to 90°. This holds across the full angular range. There is nothing special at the sector boundary or any other angle. This configuration provides a large amount of freedom for the design of the inductive sensor. Unlike accurate inductive sensors, which require large coils on the PCB (several cm), here, we can afford chip-scale coils. This size reduction yields weaker signals for the inductive sensor, significantly degrading its own accuracy, but this error is inconsequential (up to a certain relaxed limit). The unique hybrid target is composed of a metallic half-moon attached to the bottom of the four-pole magnet. The four-pole magnet is made of ferrite and, hence, is non-conductive in order to prevent disturbing the inductive sensor.

The prototype, illustrated in Figure 5, is realized with commercially available sensors: MLX90377 [14] for the magnetic part, and MLX90510 [15] for the inductive part. A coil system that is 15 mm wide has been designed. The size and turns of the transmitting coil are designed with an inductance of around 1.1 μH. This value is just enough to keep an acceptable current level and oscillation frequency for the inductive sensor, which is normally operated with larger inductance. A redesign of the IC would allow for an even smaller coil system. The coil design allows for the detection of the angular position of a half-moon metal target. The magnetic sensor is placed at the center of this coil system, such that the sensitive spot is aligned with the target for both sensors.

## 5. Measurement Setup

We focus on the analysis of the non-linearity error (NLE). The NLE is the difference between the measured angle and the actual rotation angle of the rotary stage. The NLE at room temperature can be readily calibrated out in its final operating environment. A common way to calibrate such sensors is to save the measured NLE at room temperature in the sensor memory and to subsequently subtract this saved error curve during operation. Both sensors used herein support this NLE calibration. With such a calibration, only the drifts (thermal, lifetime, etc.) impact the calibrated sensor accuracy.

The setup is shown in Figure 6. A PI Hexapod H-811.D1 controls the airgap, tilt angles and in-plane off-axis displacements. A PI DT-34 rotary stage controls the rotation angle of the hybrid target. The Hexapod is controlled from a Python script running on a laptop. Two different multimeters Keithley K2000 are connected to the two differential analog outputs of the inductive sensor, measuring sin(α) and cos(α), where α is the angular position of the metal target. These two quadrature signals are used to retrieve the measured coarse angle with the 360° range. A programmer tool (PTC-04, also commercially available) establishes the communication between the magnetic sensor and the laptop. This tool reads from the IC memory and sends the values to the PC. The communication with the PTC-04 is managed by the same Python script.

## 6. Experimental Tests and Results

### 6.1. Nominal Behavior

We start with the proof of concept for the hybrid sensor. This test was performed in nominal working conditions (airgap = 2 mm): no mechanical misalignment, no stray field and at room temperature. Two sets of measurements have been taken: one with and one without the metal target. The results in Figure 7 show that the hybrid sensor is working as expected. The range is extended up to 360° by using the measurements of the inductive sensor. The inductive sensor’s presence does not affect the magnetic sensor’s behaviour: the angular error of the magnetic sensor is the same with or without the inductive part of the hybrid sensor. The difference between the curves (inset of Figure 7) shows a random variation around 0° of up to ±0.25 independent of the angular position of the target and consistent with the expected noise level of the magnetic sensor.

### 6.2. Stray-Field Immunity

In this test, the sensor is subjected to the presence of an external constant magnetic field, and the NLE is measured as before. Figure 8 shows the error curves for both sensors. Note that the stray field is generated by a neodymium magnet placed nearby (inset of Figure 8), but is still sufficiently far such that the stray field over the sensor can be considered uniform, and so it is mostly rejected by the differential operation of the magnetic sensor. The magnetic field lines from this parasitic magnet are mostly parallel to the chip plane, which is the worst case. The amplitude of the stray magnetic field has been measured using a SENIS Teslameter whose probe has been placed right on top of the magnetic sensor. The total amplitude of the stray field applied on the sensor is 4.39 mT.

The NLE curves follow the same trend in the presence and absence of the stray field. This demonstrates that the inductive and magnetic sensors are both largely stray-field immune. The magnetic sensor NLE curve was remarkably stable, denoting a high degree of immunity to the environment. The NLE error of the inductive sensor shows a small but noticeable shift, despite being completely stray-field-immune by nature. This shift is due to the proximity of the neodymium object, which, being a good conductor, behaves as a stray metal for the inductive sensor. The eddy currents are not only generated in the metal target but also at the surface of the neodymium magnet. This parasitic contribution to the inductive three-phase sensed signals is independent of the target rotation. Mathematically, it behaves as a fixed error in the three-phase signals (instead of the ideal three-phase signals {a,b,c}, the individual phases are offsets $\left\{a+\epsilon_{a},b+\epsilon_{b},c+\epsilon_{c}\right\}$). In the angular domain, the net effect of this stray metal shows up as a first-order harmonic in the NLE, with one period per 360°.

### 6.3. Impact of Mechanical Misalignments

All tests so far were conducted in the ideal mechanical configuration with the hybrid target centered with respect to the magnetic sensor and the coil system. Now, we study the impact of mechanical misalignments. Three types of misalignments are considered: in-plane misalignment (so called off-axis displacement), airgap variation and tilt.

First, we evaluate the impact of off-axis. In this test, the center of the hybrid target is shifted by 1 mm off-axis with respect to the sensor sensing center. The resulting errors are shown in Figure 9. The magnetic sensor exhibits a first harmonic error, which increases with the off-axis displacement. In particular, for each millimeter of off-axis, the NLE increases by 2°. A similar first harmonic error is also present for the inductive sensor, but the effect is higher: 4°/mm. The magnetic sensor benefits from the error compensation due to its periodicity.

Next, we evaluate the impact of airgap variation. In this test, the vertical position of the hybrid target varies while maintaining the target’s center and the sensor’s sensitive spot as aligned. For both sensors, the impact of airgap variation is modest, as seen in Figure 10. For the inductive sensor, the NLE increases by about 0.4° for each mm of air gap change. This is expected, as variations in the airgap preserve the symmetry. These variations have an impact on the signal strength, and hence on the signal-to-noise ratio. Hence, there is a significant impact on the angular noise but not on the NLE.

Then, we evaluate the impact of tilt. The impact of tilt is different for both sensors. For the magnetic sensor, the net effect is limited thanks to the 180° periodicity of the magnet and sensor. Intuitively, a tilt brings one side of the magnet and its related poles closer to the sensor, but, at the same time, there are equivalent poles on the diametrically opposite side of the magnet, which are pushed farther away from the sensor. The contributions of the two magnet sides are summed up and averaged by the magnetic sensor for a zero net effect. The 360° inductive sensor with the half-moon target responds differently, on the other hand. A tilt will, in general, push one part of the target and the related surface eddy currents closer to the coils, whereas another part is pushed further away. However, contrary to the magnet, there is not a diametrically opposite eddy current distribution that can compensate for the tilt effect. As illustrated in Figure 11, this will result in a sinusoidal error curve. Note, however, that this is *not* an intrinsic technology limitation. As discussed previously, for inductive designs based on multi-lobe targets and accordingly adapted coil structures, the system self-compensates for the tilt effects like for the magnetic case. The only exceptions are the 360° designs. The different behaviors are clearly visible in Figure 11. The inductive sensor is much more sensitive to tilt: its NLE increases by 3.5° per degree of tilt (with the error shape being a first harmonic). The magnetic sensor curves remain within its noise band.

Finally, we evaluate the overall impact of mechanical misalignments. As expected, the inductive sensor is overall more affected than the magnetic sensor. As explained in the previous sections, this is due to the fact that the inductive sensor is designed for a 360° range. However, given the relaxed specification limits on the inductive sensor, this is inconsequential. The hybrid sensor, as a whole, retains the mechanical robustness of the magnetic sensor. Table 2 summarizes the misalignment impacts.

### 6.4. Impact of Stray Metals

As explained earlier, stray metals impact the inductive sensor due to the extra parasitic eddy currents induced on their surface. Again, given the relaxed limit, we expect the hybrid sensor to be tolerant to substantial interference. Our experiment is depicted in Figure 12. We placed aluminum foils near or even overlapping the inductive coil system. In a regular accurate inductive sensor, there would be strict design rules constraining the placement of nearby metallic structures (including PCB tracks). Here, we emulate a case where such design rules are completely absent and the hybrid sensor can be treated as a pure magnetic sensor that is largely insensitive to the metallic environment. Unsurprisingly, the inductive error curve in Figure 12 shows a significant deviation (48.42°) given the polluted metallic environment. Nevertheless, the deviation is still within the relaxed specification limit. The hybrid sensor accuracy is still defined by the magnetic sensor, and not compromised at all by the presence of stray metals.

## 7. Miniaturization Potential

In this section, we clarify the challenges to further miniaturize the presented concept. The main effort is related to the inductive sensor, whose coil sysem dominates the footprint. Regular inductive sensors require a coil system of several tens of millimeters on the PCB. The dimension is imposed by the fact that the signal captured by the receiving coils drops exponentially with a higher airgap and smaller coil diameter. Therefore, small inductive sensors are characterized by a low accuracy. On the other hand, in the context of this hybrid approach, the accuracy of the inductive sensor is not critical, and the coil diameter could be scaled down.

To study the physical limits of the inductive sensor design, we realized a chip-scale coil system (5 mm × 5 mm). Figure 13 shows the measurement result from this design, together with a picture of the coil system. The mechanical and wiring asymmetries (in the coil design and/or the alignment) naturally have a larger relative impact. For example, the asymmetries related to the feeding wires and vias are magnified in such miniaturized coil system. Still, the error remains low enough for this sensor to discriminate between the two sectors. Therefore, a hybrid sensor relying on this miniaturized chip-scale inductive sensor would retain the same range and accuracy as the hybrid sensor presented in this paper.

## 8. Conclusions

By combining the inductive and magnetic sensing principles, we have obtained a hybrid sensor. It is free of the usual range-vs.-accuracy trade-off. We have demonstrated that the two sensing paths are largely independent, allowing for straightforward design optimization. The magnetic sensor can be optimized for accuracy on its own, while the inductive part can tolerate large errors. Hence, the inductive part can be optimized for cost and made much smaller than traditional inductive sensors. Finally, the overall power consumption can be reduced by optimizing the resonance circuit for the transmitting coil. The PCB footprint and the design constraints on the metallic environment are minimized. In this work, where the inductive IC was used as-is, the inductive coil system still required a small PCB. With an IC redesign, a chip-scale implementation can be envisioned thanks to the relaxed specification.

To put our results in perspective, Table 3 compares our hybrid sensor prototype to other commercial automotive sensors (including the individual sensors used for the hybrid) in terms of tolerable stray field (“Tol. SF”), range, accuracy, mechanical error, dimensions and current consumption. The table demonstrates the improved range-vs.-accuracy trade-off while being competitive in the other metrics, apart from the current consumption, which is penalized by the need to operate two readout chains. With a chip redesign of the inductive sensor, the extra current consumption could be mitigated.

More generally, our hybrid approach opens new design options for combining magnetic and inductive technologies to address different specs. For example, a complementary hybrid sensor could be envisioned with the inductive and magnetic roles reversed. The inductive part would provide the accurate channel, with a coarse magnetic sensor as a support for the 360° range. This arrangement would benefit from the improved thermal drift in inductive technologies while maintaining the range.

In conclusion, our work opens the possibilities for a variety of hybrid concepts, which can bring improvements in accuracy without sacrificing the sensing range.

## Figures and Tables

**Figure 1 sensors-22-02153-f001:**
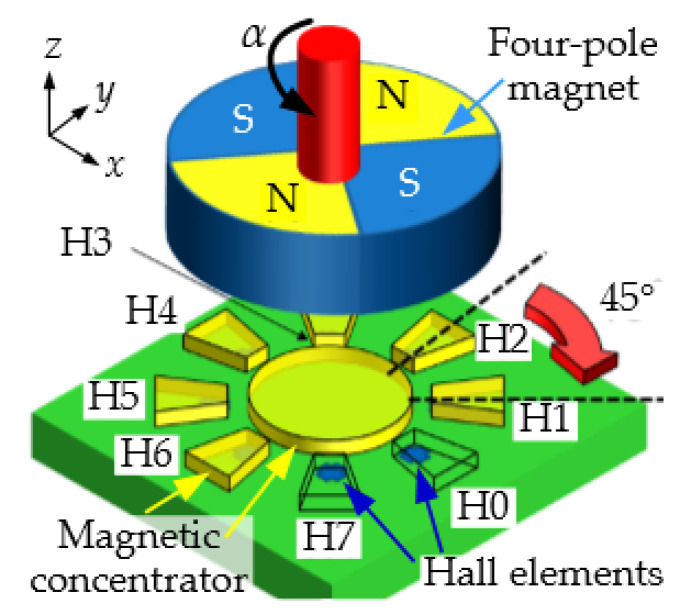
Concept diagram of the magnetic sensor and 4-pole magnet. The sensor consist of an array of 8 Hall elements covered by a magneto-concentrator structure. This arrangement is sensitive to the field gradient in the XY plane. Uniform stray fields are rejected. The figure is extracted from [8] (license CC BY 4.0).

**Figure 2 sensors-22-02153-f002:**
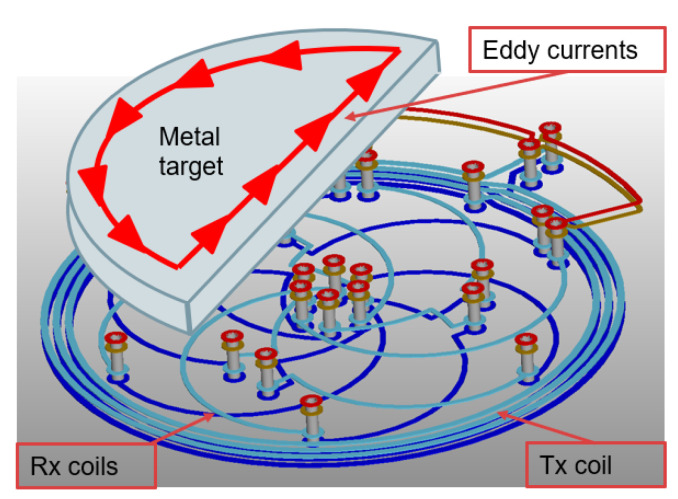
Concept diagram of the inductive sensor and target. The sensor consists of a circular transmitting (Tx) coil and a set of three receiving coils. The TX coil is excited by a periodic signal, thereby inducing eddy currents in the target. The receiving (Rx) coils sense a three-phase signal representative of the target angle.

**Figure 3 sensors-22-02153-f003:**
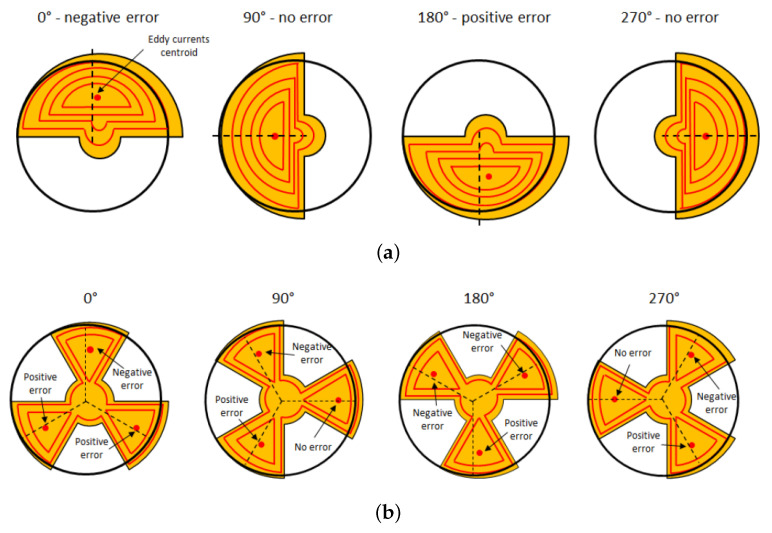
Centroid of the eddy currents for a slight off-axis shift for 4 rotation angles. The dotted line represents the ideal angular position. (**a**) Case of a 360°-capable sensor (periodicity = 1) with a half-moon target. The centroid is oscillating around its ideal rotating position, introducing a periodic error. (**b**) Case of a 120°-periodic sensor (periodicity = 3) with a three-lobe target. The three centroids are again oscillating, but there is a built-in compensation.

**Figure 4 sensors-22-02153-f004:**
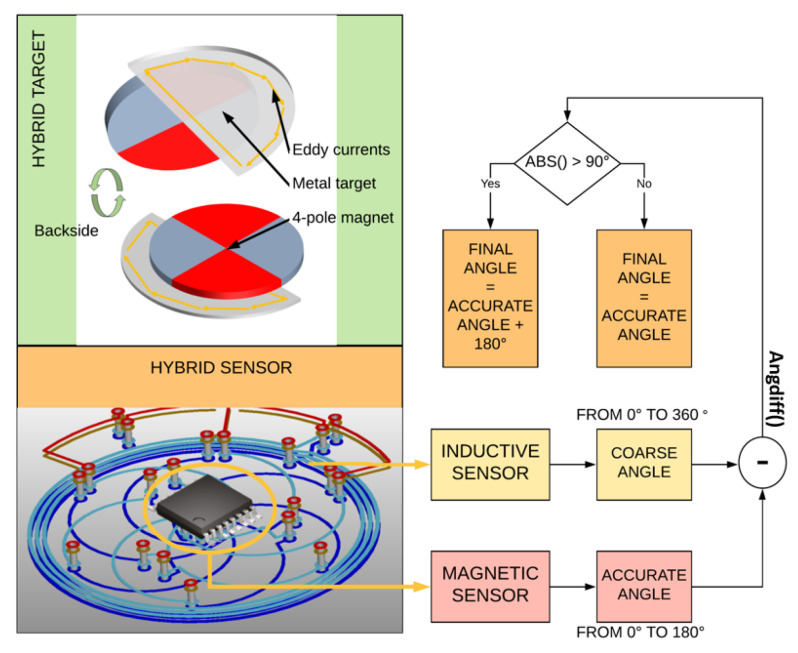
Concept diagram of the hybrid sensor and target. The hybrid sensor consists of a magnetic sensor providing an accurate angle in the 180° range, an inductive sensor providing a coarse angle and an algorithm combining both angles into a final angle. The target combines a metallic half-moon disk and a 4-pole ferrite magnet. Refer to Equation (Equation 1) for a mathematical description of the hybrid angle combination.

**Figure 5 sensors-22-02153-f005:**
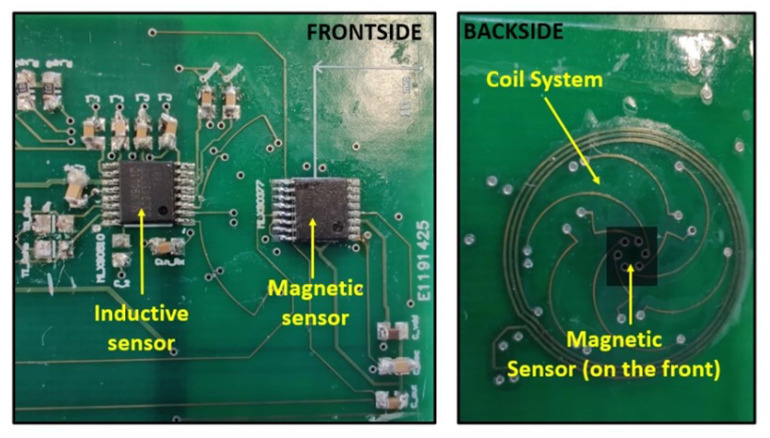
Pictures of the hybrid sensor prototype. The front side (**left** picture) of the PCB contains the two sensor ICs (magnetic and inductive). The back side (**right** picture) contains the coil system. The magnetic sensor is placed at the center of the coil system.

**Figure 6 sensors-22-02153-f006:**
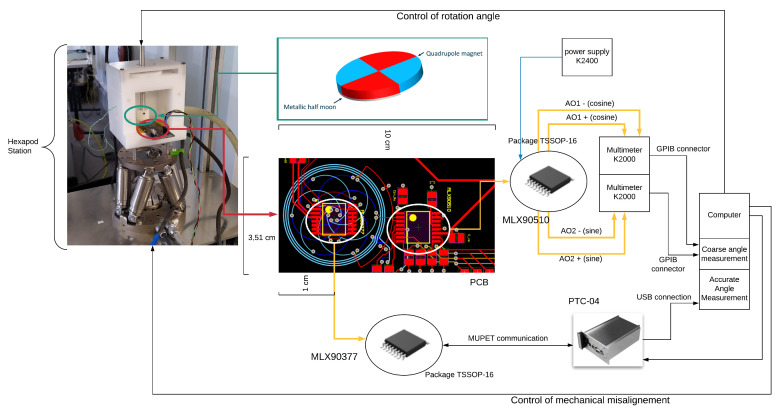
Block diagram of the hybrid sensor embedded in the measurement setup.

**Figure 7 sensors-22-02153-f007:**
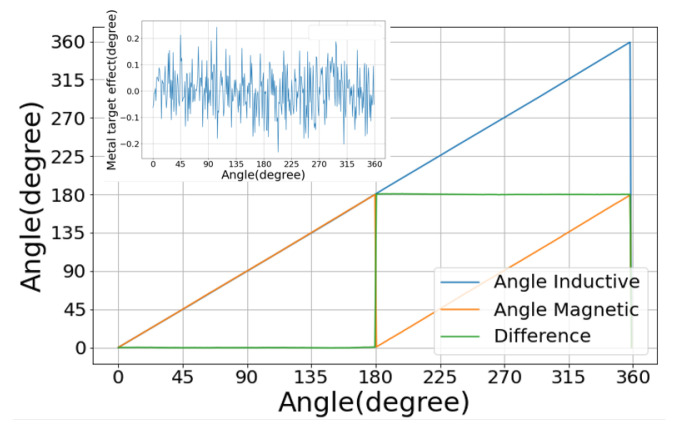
Angle response curves of the magnetic and inductive sensors. The inset shows the difference between the two magnetic measurements: with and without the metallic target. The difference is random, indicating that the magnetic sensor accuracy is not impacted by the metallic target.

**Figure 8 sensors-22-02153-f008:**
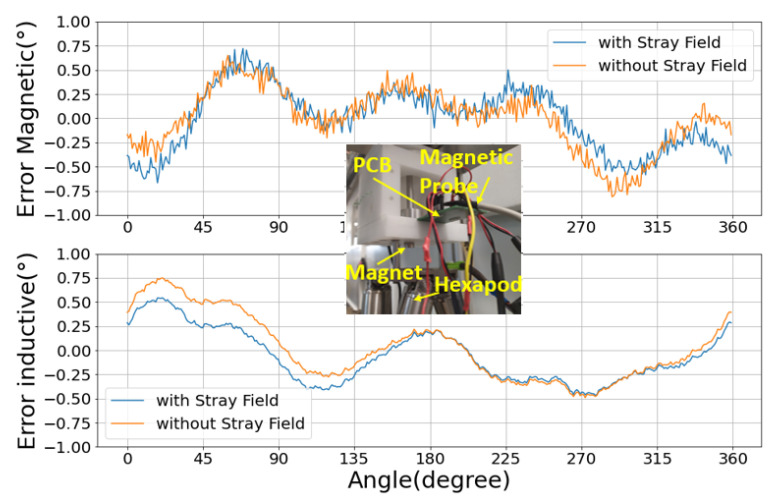
Angle response curves with and without a magnetic stray field generated by a nearby parasitic magnet. The inset is a picture of the experimental configuration. The stray field impact is modest for both sensors.

**Figure 9 sensors-22-02153-f009:**
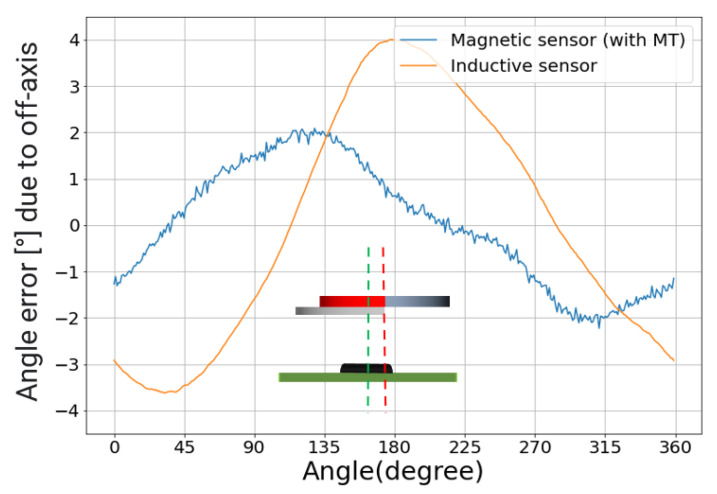
Angle error curves due to 1 mm of off-axis shift of the target. The impact is twice as large on the inductive sensor. The inset depicts the configuration. The dotted green line is the center line through the sensor center. The dotted red line is the center line through the target center.

**Figure 10 sensors-22-02153-f010:**
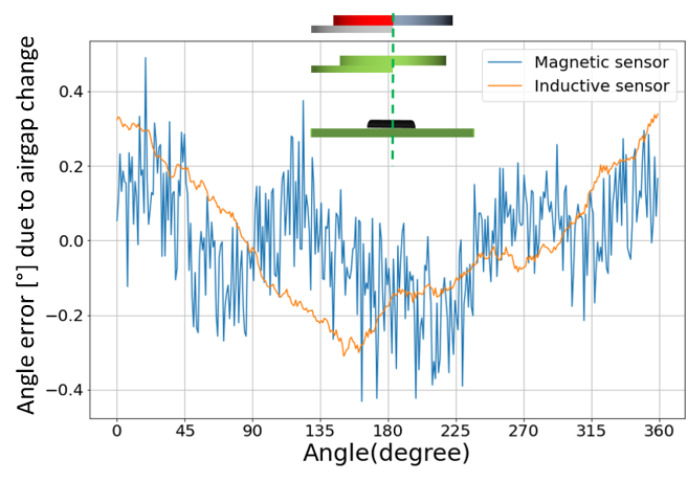
Angle error curves due to a 1 mm airgap shift. The inset depicts the configuration. The dotted green line is the center line through the sensor and target centers (which remain aligned in this test). The impact is modest.

**Figure 11 sensors-22-02153-f011:**
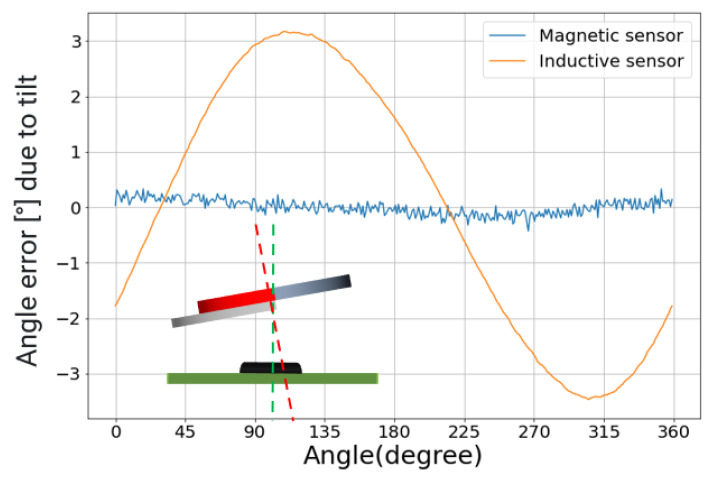
Angle error curves due to 1° of tilt. The impact is much larger on the inductive sensor. The inset depicts the configuration. The dotted green line is the center line through the sensor center. The dotted red line is the center line through the target center.

**Figure 12 sensors-22-02153-f012:**
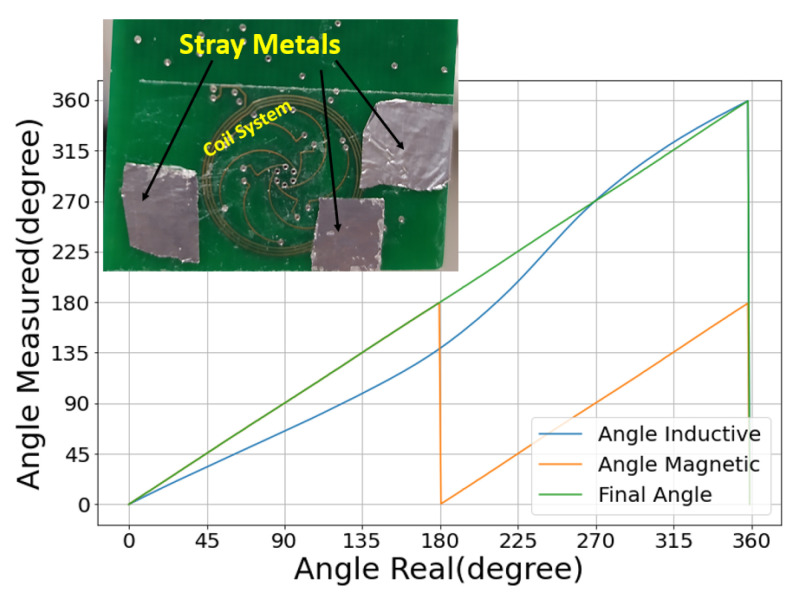
Angle response curves in the presence of stray metals. The inset is a picture of the experimental configuration showing the stray metals near or even overlapping the coil system. This pollutes the inductive sensor, yielding a substantial shift in the angle response curve of the inductive sensor. The final angle, however, is not compromised.

**Figure 13 sensors-22-02153-f013:**
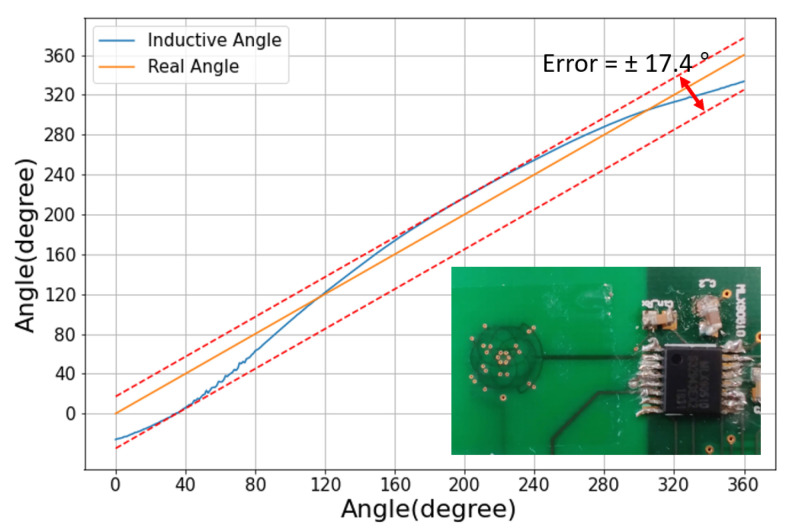
Angle response curves for the chip-scale miniaturized inductive sensor. The inset shows a picture of the chip-scale coil system and the interface chip in a TSSOP16 package.

**Table 1 sensors-22-02153-t001:** Target requirements.

Quantity	Target	Comment
Tolerable stray field	5 mT	Source: most stringent automotive norm [9]
Range	360°	Symmetrical targets are excluded (e.g., three-lobe metallic target, four-pole magnet)
Accuracy	0.6°	Also includes the temperature and lifetime drifts
Mechanical error	2°/mm	Non-linearity error increase per displacement
Dimensions	Chip scale	No large (cm scale) external components on PCB

**Table 2 sensors-22-02153-t002:** Summary of the errors due to mechanical misalignment.

Sensor	Errors			
	X Off-Axis	Y Off-Axis	Tilt	Airgap
magnetic sensor	2°@1 mm	2°@1 mm	Unaffected	More noise
inductive sensor	4°@1 mm	4°@1 mm	3.5°@1°	0.4°@ 1 mm

**Table 3 sensors-22-02153-t003:** Comparison with other existing automotive position sensors. Note that “N” is the periodicity of the inductive sensor. The dimensions do not take into account the target.

Sensors	Tol. SF (mT)	Range (°)	Accuracy (°)	Mech. Error	Dim. (mm)/ Package	Current
This work	5	360	0.67°	2°/mm	16 × 16 (Coils)	26 mA
MLX90377 [14]	5	180	0.67°	low	TSSOP16	9 mA
MLX90510 [15] (N = 1)	*∞*	360	0.2°	4°/mm	40 × 40 (Coils)	10 mA
MLX90510 [15] (N > 1)	*∞*	360/N	0.2°/N	0.2°/mm	40 × 40 (Coils)	10 mA
AS5171 [16]	<5	360	0.9°	-	SIP	5 mA
KMZ41 [17]	≪1 *	180	0.25°	-	SOIC8	-

(*) Estimated value. No SF rejection statement in the datasheet.

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
