# Peer review of "Hybrid Magnetic–Inductive Angular Sensor with 360° Range and Stray-Field Immunity"

_sensors, 2022, doi:10.3390/s22062153_

Round 1

Reviewer 1 Report

This manuscript proposed an angular sensing system with both high precision and stray-field immunity based on the integration of two different sensors. Their designs are both feasible and beneficial in engineering applications. However, the scientific issues and the manuscript organization still need to be improved.

1.Authors should better clarify their application scenarios and present the corresponding demonstration experiments.

2.The proposed sensing technology needs to determine the sector of the target first rely on a low-precision inductive sensor. Due to the relatively large angular range of a sector, this procedure is usually reliable. But what will happen if the target is near the boundary of the two sectors?

3.The specific configuration of this integration problem needs to be optimized. The fundamental goal of this work is to achieve higher precision detection in 360 degrees within a limited size. Thus, for example, increasing the order of the magnetic sensor and increasing the size of the inductive sensor are both beneficial. But, limited by the total size, there must be a trade-off between these two parameters to achieve the best cooperative detection performance.

4.Perhaps, under different interference strength and different total size constraints, the optimal design solutions of the above integration problems are also different.

5.In terms of prototyping design, is it possible to further reduce the overall size of the system through a structured design of the two sensors and other components?

6.If the proposed sensing technology is for in-vehicle applications, other types of detection techniques should also be summarized in the introduction.

Ref:

Hua T, Dai K, Zhang X, et al. Optimal VMD-based signal denoising for laser radar via Hausdorff distance and wavelet transform[J]. IEEE Access, 2019, 7: 167997-168010.

Qian K, Zhu S, Zhang X, et al. Robust Multimodal Vehicle Detection in Foggy Weather Using Complementary Lidar and Radar Signals[C]//Proceedings of the IEEE/CVF Conference on Computer Vision and Pattern Recognition. 2021: 444-453.
Son Y S, Sung H K, Heo S W. Automotive frequency modulated continuous wave radar interference reduction using per-vehicle chirp sequences[J]. Sensors, 2018, 18(9): 2831.

Author Response

Point 1. Authors should better clarify their application scenarios and present the corresponding demonstration experiments.

We added a list of typical applications (“accelerator and brake pedals, throttle valves, ride height sensors, turbo actuators, coolant valves, steering angle sensors, parking pawl sensors and motor position sensors …”) in the introduction. We also clarified the need for 360° by discussing the example of coolant valves prevalent in electric vehicles. To support this case, we added the following reference:

  • Li, B.; Kuo, H.; Wang, X.; Chen, Y.; Wang, Y.; Gerada, D.; Worall, S.; Stone, I.; Yan, Y. Thermal Management of Electrified Propulsion System for Low-Carbon Vehicles. Automot. Innov. 2020, 3, 299–316. doi:10.1007/s42154-020-00124-y.

Point 2. The proposed sensing technology needs to determine the sector of the target first rely on a low-precision inductive sensor. Due to the relatively large angular range of a sector, this procedure is usually reliable. But what will happen if the target is near the boundary of the two sectors?

We added equation (1) and the companion mathematical explanation to clarify that there is no special angle or boundaries where the sensor behavior would change. In a nutshell, the inductive angle is not used directly in an absolute comparison with 180°. Instead, only its deviation with respect to the magnetic angle is used. The error behavior is then uniform throughout the angle range and there is no special or boundary angle. 

Point 3.The specific configuration of this integration problem needs to be optimized. The fundamental goal of this work is to achieve higher precision detection in 360 degrees within a limited size. Thus, for example, increasing the order of the magnetic sensor and increasing the size of the inductive sensor are both beneficial. But, limited by the total size, there must be a trade-off between these two parameters to achieve the best cooperative detection performance.

We added a trade-off discussion related to the high-order magnetic system (6-pole magnet, and beyond). We explained that increasing the order beyond the 4-pole magnet is sub-optimal, thereby justifying our configuration.

Point 4. Perhaps, under different interference strength and different total size constraints, the optimal design solutions of the above integration problems are also different.

This point is indeed true. Under very different circumstances, another optimum exists. This is then a completely different concept where the sensor is on purpose placed off-axis under the poles of a multi-pole magnets with up to 32 pole pairs. Such configurations brings extra constraints (cost, mechanical assembly …) limiting its deployment to niche applications (e.g. robotics). We added the following reference to acknowledge this different configuration:

  • iC Haus. iC-MU - Magnetic Off-Axis Position Encoder. https://www.ichaus.de/upload/pdf/MU_datasheet_F2en.pdf, accessed: 25-Feb-2022.

Point 5. In terms of prototyping design, is it possible to further reduce the overall size of the system through a structured design of the two sensors and other components? 

We added a section titled “Miniaturization potential” to address this point. The new section focuses on the miniaturization of the inductive sensor as this dominates the footprint. We actually realized such miniaturized design. We added Fig. 13 to show this design and its performance.

Point 6. If the proposed sensing technology is for in-vehicle applications, other types of detection techniques should also be summarized in the introduction.

We clarified the focus already in response to point 1. The focus is on sensors embedded in mechatronic vehicle sub-systems (pedals, valves …). Note that the two references suggested describe radar systems to measure vehicle-to-vehicle distance. They are then out of scope. 

Reviewer 2 Report

In this article, Brajon et al. propose a new concept for angular sensing based on the synergy of magnetic and inductive sensors. The magnetic sensing is carried out by a triaxial Hall sensor IC based on distributed Hall elements and magnetic flux concentrators, and the inductive sensing by a pair of transmitter and receiver coils, detecting the changes in magnetic field due to the eddy currents in metallic targets. The authors present and evaluate a prototype of this idea, with respect to its angular accuracy and resistance to positional, magnetic and metallic disturbances. The combination of both types of sensing could indeed bring benefits for angular detection by improving the redundancy and robustness of the system. Improving these parameters might prove important for automotive sensing, where electromagnetic and mechanical disturbances are very common. 

  • General concept comments

The manuscript is clear, well-structured, and easy to follow for most of the sections. The main line of argumentation of the authors is that a novel kind of sensor needs to be developed to comply with the latest SFI requirements. As AMR sensors have a mostly non-linear response, stray fields are particularly hard to compensate compared to the case of linear elements like Hall Effect sensors. While this assertion is true, I believe the manuscript could be strengthened by contrasting the performance of AMR sensors with the proposed solution by the authors. Even though the hybrid sensor presents improvements in SFI and mechanical robustness, it also implies an increased complexity, size, and potentially more power consumption. Furthermore, it would be interesting to still compare a standard AMR solution with the sensors in Table 3 to appreciate even more the contribution of the authors. The table could also include a power and/or cost columns, for example, which could foster a short discussion about the added benefits of the method presented. The dimensions column could provide more quantitative data like footprint, area, or volume in some units, otherwise it is hard to judge how big a small PCB is. One could argue that AMR sensors, with their intrinsic advantages, can still be used if proper shielding or AC excitation are implemented. Is this path viable? This work can be made even more solid if the authors discuss the trade-offs involved in following the AMR path.

  • Specific comments

The abstract can be improved by explicitly stating what is meant in 2 sentences:

Sentence 1 --> … at the same time enables superior performances …, here the text after superior performances should be replaced by the explicit parameters that are improved.

Sentence 2 --> … demonstrate the orthogonality … beyond the usual trade-offs, here orthogonality is confusing and could be replaced by complementarity or similar. Please state explicitly what is meant with usual trade-offs.

Figures 9, 10 and 11 should have explicit units on the y axes.

Author Response

Point 1. Furthermore, it would be interesting to still compare a standard AMR solution with the sensors in Table 3 to appreciate even more the contribution of the authors. 

We added the following AMR sensor to the comparison table:

  • NXP. KMZ41 - AMR Programmable angle sensor. https://www.nxp.com/docs/en/data-sheet/KMZ41.pdf, accessed: 21-Feb-2022.

Indeed such sensor is attractive from an accuracy and size standpoint. The issue, as mentioned in the introduction, is the lack of stray-field immunity. It can only tolerate stray field << 1mT.

Point 2. The table could also include a power and/or cost columns, for example, which could foster a short discussion about the added benefits of the method presented. 

We added the current consumption to the comparison table. The current consumption is a downside of our approach which is is penalized by the need to operate two readout chains. We acknowledge this point explicitly in a new caveat we added in the conclusions.

Point 3. The dimensions column could provide more quantitative data like footprint, area, or volume in some units, otherwise it is hard to judge how big a “small PCB” is.

We added the PCB dimensions for the inductive sensors explicitly in the comparison table (X by Y mm) in the table.

Point 4.  One could argue that AMR sensors, with their intrinsic advantages, can still be used if proper shielding or AC excitation are implemented. Is this path viable? This work can be made even more solid if the authors discuss the trade-offs involved in following the AMR path.

We acknowledged shielding as a viable, but bulky, option. We added the following reference to justify this statement. 

  • A. Faria, J. Fontainhas, D. Araujo, J. Cabral and L. A. Rocha, "Study of the shielding of angular position sensors with magnetic transduction," 2016 17th International Conference on Thermal, Mechanical and Multi-Physics Simulation and Experiments in Microelectronics and Microsystems (EuroSimE), 2016, pp. 1-5, doi: 10.1109/EuroSimE.2016.7463330.

AC excitation is not an option for magnetic sensors, as far as we know. AC excitation would require to AC modulate the magnet itself. This is not possible. 

Point 5. The abstract can be improved by explicitly stating what is meant in 2 sentences:

Sentence 1 --> ... at the same time enables superior performances ..., here the text after superior performances should be replaced by the explicit parameters that are improved. Sentence 2 --> ... demonstrate the orthogonality ... beyond the usual trade-offs, here orthogonality is confusing and could be replaced by complementary or similar. Please state explicitly what is meant with usual trade-offs.

We modified the two sentences as suggested (changes are in italic):

  • “Using the two principles at the same time enables superior performances, in terms of range, compactness and robustness.
  • "The measurement results demonstrate the two sensing principles are completely independent, thereby opening the doors for hybrid optimum magnetic-inductive designs beyond the usual trade-offs (range vs resolution, size vs robustness to misalignment)."

Point 6. Figures 9, 10 and 11 should have explicit units on the y axes.

We added the unit (°) to these three graphs.

Round 2

Reviewer 1 Report

The revised manuscript has well addressed my concerns.